# Implicit Regularization of Bregman Proximal Point Algorithm and Mirror Descent on Separable Data

## Abstract

Bregman proximal point algorithm (BPPA), as one of the centerpieces in the optimization toolbox, has been witnessing emerging applications. With simple and easy to implement update rule, the algorithm bears several compelling intuitions for empirical successes, yet rigorous justifications are still largely unexplored. We study the computational properties of BPPA through classification tasks with separable data, and demonstrate provable algorithmic regularization effects associated with BPPA. We show that BPPA attains non-trivial margin, which closely depends on the condition number of the distance generating function inducing the Bregman divergence. We further demonstrate that the dependence on the condition number is tight for a class of problems, thus showing the importance of divergence in affecting the quality of the obtained solutions. In addition, we extend our findings to mirror descent (MD), for which we establish similar connections between the margin and Bregman divergence. We demonstrate through a concrete example, and show BPPA/MD converges in direction to the maximal margin solution with respect to the Mahalanobis distance. Our theoretical findings are among the first to demonstrate the benign learning properties BPPA/MD, and also provide strong corroborations for a careful choice of divergence in the algorithmic design.

## 1 Introduction

The role of optimization algorithms has become arguably one of the most critical factors in the empirical successes of training deep models. As the go-to choice for modern machine learning, first-order algorithms, including (stochastic) gradient descent and their adaptive counterparts (Kingma and Ba, 2014; Duchi et al., 2011), have received tremendous attention, with detailed investigations dedicated to understanding the effect of batch size (Goyal et al., 2017; Smith et al., 2018; Keskar et al., 2016), learning rate (Li et al., 2019; He et al., 2019; Lewkowycz et al., 2020), and momentum (Sutskever et al., 2013; Smith, 2018) across a broad spectrum of applications.

Meanwhile, Bregman proximal point algorithm (Eckstein, 1993; Kiwiel, 1997) has been drawing substantial interests. The resounding successes of this classical algorithm are particularly evident for applications including knowledge distillation (Furlanello et al., 2018), mean-teacher learning paradigm (Tarvainen and Valpola, 2017), few-shot learning (Zhou et al., 2019), policy optimization (Green et al., 2019), and fine-tuning pre-trained models (Jiang et al., 2020), yielding competitive performance compared to its first-order counterparts. In the general form, Bregman proximal point algorithm updates parameters by minimizing a loss $L(\cdot)$, while regularizing the weighted distance to the previous iterate measured by some divergence function $\mathcal{D}(\cdot, \cdot)$,

$$\theta_{t+1} = \operatorname{argmin}_\theta L(\theta) + 1/(2\eta_t)\mathcal{D}(\theta, \theta_t). \tag{1.1}$$

Popular choices of divergence function used in practice include the squared $\ell_2$-norm distance $D_{\mathrm{LS}}(\theta, \theta_t) = \mathbb{E}_\mathcal{D} \|f_\theta(x) - f_{\theta_t}(x)\|_2^2$ (Tarvainen and Valpola, 2017), and Kullback-Leibler based divergence $D_{\mathrm{KL}}(\theta, \theta_t) = \mathbb{E}_\mathcal{D}\mathrm{KL}\left(f_{\theta'}(x)\|f_\theta(x)\right)$ (Furlanello et al., 2018), where $\mathcal{D}$ denotes the data distribution. Such a simple update is of great practical purposes, as it is easy to describe, and admits simple implementation by adopting suitable off-the-shelf black-box optimization algorithms (Solodov and Svaiter, 2000; Monteiro and Svaiter, 2010; Zaslavski, 2010). The updating form also suggests plausible intuitions for its empirical successes, including iteratively constraining the search space, alleviating aggressive updates, and preventing catastrophic forgetting (Schulman et al., 2015; Li and Hoiem, 2017). However, none of the intuitions have been rigorously justified, and

theoretical understandings for the empirical successes of Bregman proximal point algorithm remains underexplored.

A first, and a natural question is whether Bregman proximal point algorithm benefits from the same kind of mechanism that (stochastic) gradient descent (GD/SGD) enjoys for having the generalization properties. In particular, in many important applications, GD/SGD is widely believed as the "the algorithm that finds the right kind of solutions" for problems with non-unique solutions. Such a claim is supported with numerous provable examples: GD/SGD converges to the minimum-norm solution of under-determined linear systems (Gunasekar et al., 2018), converges to the max-margin solution for separable data (Soudry et al., 2018; Nacson et al., 2019), aligns layers of deep linear networks (Ji and Telgarsky, 2018), and converges to a generalizable solution for nonlinear networks (Brutzkus et al., 2017; Allen-Zhu et al., 2018) in the presence of infinitely many overfitting solutions. Given the emerging successes of Bregman proximal point algorithm, and the aforementioned evidences on its first-order counterparts (e.g. GD/SGD) finding generalizable solutions, one would naturally wonder

*Does Bregman proximal point algorithm converge to a solution with favorable qualities?*

Another important question with great practical implications for Bregman proximal point algorithm is how the divergence measure $\mathcal{D}(\cdot, \cdot)$ affects the solution. Instead of directly applying the Euclidean distance based divergence, it is widely observed that the successful application of Bregman proximal point algorithm is contingent on the careful design of divergence measure, based on the task at hand (Li and Hoiem, 2017; Hinton et al., 2015). Take the example of fine-tuning language model, the symmetrized Kullback-Leibler based divergence evaluated on the predictions of the updated model (i.e., $\theta_{t+1}$) and previous model (i.e., $\theta_t$) yields the state-of-the-art result (Jiang et al., 2020). Identifying the underlying mechanism for the success or failure of a given divergence choice is not only of theoretical interest, but also can significantly reduce human effort in searching/designing the suitable divergence for a given task. As an important addition, one may also ask whether the impact of divergence on the Bregman proximal point algorithm find natural counterparts in commonly adopted first-order algorithms (e.g. mirror descent, (Nemirovski and Yudin, 1983)). In such cases, better task-dependent algorithmic designs could be proposed in conjunction with the suitable divergence. To this end, we raise our second question.

*How does divergence affect the qualities of the solution obtained by Bregman proximal point algorithm (and other first-order algorithms)?*

In this paper, we initiate our study to address our previously proposed questions. We focus on a non-trivial example of an under-determined system – training linear classifiers using separable data. In particular, for exponential tail losses (e.g., exponential loss), the empirical loss function has infimum zero that is asymptotically attainable at infinity along infinitely many directions. The natural candidate for measuring the quality of the obtained classifier is its margin, i.e., the minimum distance between the samples and the decision hyperplane. For such a problem, we summarize our theoretical findings below as concrete answers to the previous questions.

- We show that Bregman proximal point algorithm (BPPA) obtains a solution with non-trivial margin lower-bound. As a concrete demonstration, we tailor our main theorem for Mahalanobis distance, and show that BPPA converges in direction to the maximal margin solution. We provide non-asymptotic analyses of the margin and empirical loss for constant stepsize BPPA, and propose a more aggressive stepsize rule for a provable exponential speed-up.
- We establish a dependence of such a margin lower-bound on the condition number of the distance generating function for defining the divergence. In addition, we provide a class of problems where the margin lower-bound is tight, demonstrating that the Bregman divergence is crucial in affecting the quality of the obtained solution.
- We extend our findings to first-order algorithms. Specifically, we show that mirror descent (MD) enjoys the same previously mentioned margin properties. We also provide non-asymptotic convergence analyses of the margin and empirical loss for constant stepsize MD, and its exponential speed-up using a varying stepsize scheme. Our findings for MD strictly complement prior works on under-determined regression problems (Gunasekar et al., 2018; Azizan and Hassibi, 2019).

**Notations**. We denote $[n] := \{1, \ldots, n\}$; $\text{sgn}(z) = 1$ if $z \geq 0$ and $-1$ elsewhere. We use w.r.t in short for "with respect to". For any $\|\cdot\|$ in Euclidean space $\mathbb{R}^d$, we use $\|\cdot\|_* = \max_{\|y\| \leq 1} \langle \cdot, y \rangle$ to denote its dual norm. Note that we have $(\|\cdot\|_*)_* = \|\cdot\|$.

## 2 PROBLEM SETUP

We study the binary classification on linearly separable data. Specifically, the dataset is $\mathcal{S} = \{(x_i, y_i)\}_{i=1}^n \subset \mathbb{R}^d \times \{+1, -1\}$, where $x_i$ is the feature vector, and $y_i$ is the label. In addition, there exists a linear classifier $u \in \mathbb{R}^d$ such that $y_i \langle u, x_i \rangle > 0$ for all $i \in [n]$. That is, the decision rule $f_u(\cdot) = \text{sgn}(\langle u, \cdot \rangle)$ achieves the perfect accuracy on the dataset, with $y_i = f_u(x_i)$ for all $i \in [n]$.

For each linear classifier $f_u(\cdot)$ with perfect accuracy, we define its $\|\cdot\|_*$-norm margin as the minimum distance in $\|\cdot\|_*$-norm from the feature vectors to the decision boundary $\mathcal{H}_u = \{x : \langle x, u \rangle = 0\}$. It is well known that the $\|\cdot\|_*$-norm margin, denoted as $\gamma_u$, only depends on the direction of the classifier and satisfies $\gamma_u = \min_{i \in [n]} \left\langle x_i y_i, \frac{u}{\|u\|} \right\rangle$, where $\|\cdot\|$ is the dual norm of $\|\cdot\|_*$. The $\|\cdot\|_*$-norm margin measures how well the data is separated by decision rule $f_u(\cdot)$, measured in $\|\cdot\|_*$-norm, and is an important measure on the generalizability and robustness of the decision rule. Given a norm $\|\cdot\|_*$ on $\mathbb{R}^d$, we define the optimal linear classifier with the maximum $\|\cdot\|_*$-margin below.

**Definition 2.1** (Maximum $\|\cdot\|_*$-norm Margin Classifier). *Given a linearly separable dataset* $\{(x_i, y_i)\}_{i \in [n]}$, *we define the maximum* $\|\cdot\|_*$-*norm margin classifier* $u_{\|\cdot\|_*}$, *and its associated maximum* $\|\cdot\|_*$-*norm margin* $\gamma_{\|\cdot\|_*}$ *as*

$$u_{\|\cdot\|_*} = \underset{\|u\| \leq 1}{\text{argmax}} \, \min_{i \in [n]} \langle u, y_i x_i \rangle, \quad \gamma_{\|\cdot\|_*} = \max_{\|u\| \leq 1} \min_{i \in [n]} \langle u, y_i x_i \rangle.$$

For a separable dataset, we consider finding the classifier by minimizing the empirical loss

$$L_{\mathcal{S}}(\theta) = \frac{1}{n} \sum_{i=1}^n \ell\left(\langle \theta, y_i x_i \rangle\right). \tag{2.1}$$

Here we focus on the exponential loss $\ell(x) = \exp(-x)$, and our analyses can be readily extended to other losses with tight exponential tail (e.g., logistic loss).

**Observation**. One can readily verify that with a separable dataset $\mathcal{S}$, the empirical loss has infimum 0 but possesses no finite solution that attains the infimum. Thus any optimization algorithm minimizing the loss $L_{\mathcal{S}}(\cdot)$ will observe the explosion on the norm of iterate.

It has been shown that various optimization algorithms, including (stochastic) gradient descent and steepest descent, converge in direction to the maximum margin classifier in different norms (Soudry et al., 2018; Nacson et al., 2019; Gunasekar et al., 2018; Ji and Telgarsky, 2019; 2021). Connections between gradient descent and the regularization path of homotopy method have also been established (Ji et al., 2020). A striking feature behind such phenomena is that there is no explicit regularization in the loss function, and such effects have been termed as the *implicit (algorithmic) regularization*.

Up to date, most of the implicit regularization effects are attributed to (stochastic) gradient descent, given their prevalence in applications. However, as Bregman proximal point algorithm (BPPA) becomes increasingly popular in various domains, there exists considerable lack of understanding on the computational properties of BPPA. In addition, practitioners often find the choice of divergence function crucially important for the performance of BPPA (Jiang et al., 2020; Furlanello et al., 2018). This empirical evidence thus calls for a detailed characterization on the connection between computational properties and the divergence function of BPPA.

In what follows, we study the BPPA for solving problem (2.1) in detail. The BPPA (Algorithm 1) is an adaptation of the vanilla proximal point algorithm (Rockafellar, 1976a;b) to non-euclidean geometry, by using Bregman divergence as the divergence measure in (1.1). Specifically, given a distance generating function $w(\cdot)$ that is convex and differentiable, we define the Bregman divergence $D_w(\cdot, \cdot)$ associated with $w(\cdot)$ as $D_w(\theta, \theta') = w(\theta) - w(\theta') - \langle \nabla w(\theta'), \theta - \theta' \rangle$. Throughout our discussions, we only impose the following mild assumption on Bregman divergence function $D_w(\cdot, \cdot)$.

---

**Algorithm 1** Bregman Proximal Point Algorithm (BPPA)

---

**Input:** Distance generating function $w(\cdot)$, stepsizes $\{\eta^t\}_{t \geq 0}$, samples $\{x_i, y_i\}_{i=1}^n$.
**Initialize:** $\theta^0 \leftarrow 0$.
**for** $t = 0, \ldots$ **do**
$$\text{Update } \theta_{t+1} = \underset{\theta}{\text{argmin}} \, L_{\mathcal{S}}(\theta) + \frac{1}{2\eta_t} D_w(\theta, \theta_t). \tag{2.2}$$
**end for**

---

**Assumption 1.** *We assume that the distance generating function of Bregman divergence $D_w(\cdot, \cdot)$ is $L_w$-smooth and $\mu_w$-strongly convex w.r.t. $\|\cdot\|$-norm. That is,*

$$\frac{\mu_w}{2} \|\theta - \theta'\|^2 \leq w(\theta) - w(\theta') - \langle \nabla w(\theta'), \theta - \theta' \rangle \leq \frac{L_w}{2} \|\theta - \theta'\|^2 .$$

## 3  Algorithmic Regularization of BPPA

We show BPPA achieves a $\|\cdot\|_*$-norm margin that is at least $\sqrt{\mu_w/L_w}$-fraction of the maximal one.

**Theorem 3.1** (Constant Stepsize BPPA). *Let $D_{\|\cdot\|_*} = \max_{i \in [n]} \|x_i\|_*$, where $\|\cdot\|_*$ denotes the dual norm of $\|\cdot\|$. Then under Assumption 1, for any constant stepsize $\eta_t = \eta > 0$, the following hold.*

*(1) We have $\lim_{t \to \infty} L_S(\theta_t) = 0$. Specifically, we have that $L_S(\theta_t)$ diminishes at the following rate,*

$$L_S(\theta_t) \leq \frac{1}{\gamma_{\|\cdot\|_*} \eta t} + \frac{L_w \log^2 (\gamma_{\|\cdot\|_*} \eta t)}{4 \gamma_{\|\cdot\|_*}^2 \eta t} = \mathcal{O}\left( \frac{L_w \log^2 (\gamma_{\|\cdot\|_*} \eta t)}{\gamma_{\|\cdot\|_*}^2 \eta t} \right).$$

*(2) We have that the margin is asymptotically lower bounded by*

$$\lim_{t \to \infty} \min_{i \in [n]} \left\langle \frac{\theta_t}{\|\theta_t\|}, y_i x_i \right\rangle \geq \sqrt{\frac{\mu_w}{L_w}} \gamma_{\|\cdot\|_*}, \tag{3.1}$$

*where $\gamma_{\|\cdot\|_*}$ is defined in Definition 2.1. In addition, for any given $\epsilon > 0$, there exists a $t_0$ satisfying*

$$t_0 := \widetilde{\mathcal{O}}\left( \max\left\{ \frac{D_{\|\cdot\|_*}^2}{\epsilon^2 \gamma_{\|\cdot\|_*}^2}, \exp\left( \frac{D_{\|\cdot\|_*}^2}{\gamma_{\|\cdot\|_*}^2 \epsilon^2} \sqrt{\frac{L_w}{\mu_w}} \right) \frac{1}{\gamma_{\|\cdot\|_*}^2 \eta} \right\} \right),$$

*such that for $t \geq t_0$ number of iterations, we have*

$$\left\langle \frac{\theta_t}{\|\theta_t\|}, y_i x_i \right\rangle \geq (1 - \epsilon) \sqrt{\frac{\mu_w}{L_w}} \gamma_{\|\cdot\|_*}, \quad \forall i \in [n].$$

We highlight that (1) The choice of Bregman divergence in BPPA is flexible and can be *data dependent*. Properly chosen data-dependent divergence can adapt to data geometry much better than data-independent divergence, leading to better separation and margin. In Section 5 we demonstrate how BPPA can benefit significantly from such an adaptivity of carefully designed data-dependent divergence. (2) Our analysis on the convergence requires handling non-finite minimizers, which implies divergence of iterate $\|\theta_t\| \to \infty$. The optimization problem of our interest does not meet the standard assumptions in the classical analysis of BPPA in the literature, and requires a careful choice of reference point in order to derive non-trivial convergence results. (3) Our result is closely related, but should not be confused with the homotopy method in (Rosset et al., 2004), which can be viewed as performing only one proximal step at the origin, with an extremely large stepsize. (4) Finally, our result is a generalization of Telgarsky (2013); Gunasekar et al. (2018) to non-euclidean settings with Bregman divergence. Working with Bregman divergence poses unique challenges, as it is previously unclear how to relate the primal margin progress to the per-iteration progress over the dual space.

Theorem 3.1 shows that if the distance generating function $w(\cdot)$ is well-conditioned w.r.t. $\|\cdot\|$-norm, then Bregman proximal point algorithm will output a solution with near optimal $\|\cdot\|_*$-norm margin. As a concrete realization of Theorem 3.1, we consider the Mahalanobis distance $\|\cdot\|_A := \sqrt{\langle \cdot, A \cdot \rangle}$ induced by a positive definite matrix $A$.

**Corollary 3.1.** *Let $\|\cdot\| = \|\cdot\|_A$ for some positive definite matrix $A$. Under the same conditions as in Theorem 3.1, BPPA with distance generating function $w(\cdot) = \langle \cdot, A \cdot \rangle$ converges to the maximum $\|\cdot\|_*$-margin solution, where $\|\cdot\|_* = \|\cdot\|_{A^{-1}}$. Specifically, we have*

$$L_S(\theta_t) \leq \frac{1}{\gamma_{\|\cdot\|_*} \eta t} + \frac{L_w \log^2 (\gamma_{\|\cdot\|_*} \eta t)}{4 \gamma_{\|\cdot\|_*}^2 \eta t} = \mathcal{O}\left( \frac{L_w \log^2 (\gamma_{\|\cdot\|_*} \eta t)}{\gamma_{\|\cdot\|_*}^2 \eta t} \right).$$

*In addition, we have $\lim_{t \to \infty} \min_{i \in [n]} \left\langle \frac{\theta_t}{\|\theta_t\|}, y_i x_i \right\rangle = \gamma_{\|\cdot\|_*}$. Specifically, for any given $\epsilon > 0$, there exists a $t_0$ satisfying $t_0 := \widetilde{\mathcal{O}}\left( \max\left\{ \frac{D_{\|\cdot\|_*}^2}{\epsilon^2 \gamma_{\|\cdot\|_*}^2}, \exp\left( \frac{D_{\|\cdot\|_*}^2}{\gamma_{\|\cdot\|_*}^2 \epsilon^2} \right) \frac{1}{\gamma_{\|\cdot\|_*}^2 \eta} \right\} \right)$, such that for $t \geq t_0$ number of iterations, we have*

$$\left\langle \frac{\theta_t}{\|\theta_t\|}, y_i x_i \right\rangle \geq (1 - \epsilon) \gamma_{\|\cdot\|_*}, \quad \forall i \in [n].$$

*Finally, we have the direction convergence that $\lim_{t \to \infty} \frac{\theta_t}{\|\theta_t\|} = u_{\|\cdot\|_*}$.*

Note that similar directional convergence results have been shown in Gunasekar et al. (2018) for steepest descent w.r.t. $\|\cdot\|_A$ norm. The directional convergence of BPPA obtained here, however, is not a simple corollary of known results, since existing analyses focus on first-order algorithms in euclidean setting (e.g., GD/SGD, steepest descent). Such existing analyses do not simply extend to non-first-order algorithms in non-euclidean setting, such as BPPA.

When the distance generating function $w(\cdot)$ is ill-conditioned w.r.t. $\|\cdot\|$-norm (i.e., $\sqrt{\mu_w/L_w} \ll 1$), it might be tempting to suggest that the margin lower bound in (3.1) is loose, and what really happens is $\lim_{t\to\infty} \min_{i\in[n]} \left\langle \frac{\theta_t}{\|\theta_t\|}, y_i x_i \right\rangle = \gamma_{\|\cdot\|_*}$. However, as we show in the following proposition, there exists a class of problems, where the lower bound in (3.1) is in fact a tight upper bound (up to a factor of 2), demonstrating that the dependence on condition number of distance generating function $w(\cdot)$ w.r.t. $\|\cdot\|$-norm *is not* a proof artifact.

**Proposition 3.1** (Tight Dependence on Condition Number). *There exists a sequence of problems* $\{\mathcal{P}^{(m)}\}_{m\geq 1}$*, where each* $\mathcal{P}^{(m)} = \left(\mathcal{S}^{(m)}, \|\cdot\|^{(m)}, w^{(m)}\right)$ *denotes the dataset, the norm, and the distance generating function of the m-th problem. For each m, the distance generating function* $w^{(m)}(\cdot)$ *is* $\mu_w^{(m)}$*-strongly convex and* $L_w^{(m)}$*-smooth w.r.t.* $\|\cdot\|$*-norm. Then Bregman proximal point algorithm applied to each problem in* $\{\mathcal{P}^{(m)}\}_{m\geq 1}$ *yields*

$$\lim_{t\to\infty} \min_{(x,y)\in\mathcal{S}^{(m)}} \left\langle \frac{\theta_t}{\|\theta_t\|}, yx \right\rangle \Big/ \gamma_{\|\cdot\|_*^{(m)}} \leq 2\sqrt{\frac{\mu_w^{(m)}}{L_w^{(m)}}}, \quad \forall m \geq 1, \tag{3.2}$$

*In addition, for any* $m \geq 4$*, we have* $\lim_{t\to\infty} \min_{(x,y)\in\mathcal{S}^{(m)}} \left\langle \frac{\theta_t}{\|\theta_t\|}, yx \right\rangle \Big/ \gamma_{\|\cdot\|_*^{(m)}} \leq 2\sqrt{\frac{\mu_w^{(m)}}{L_w^{(m)}}} < 1.$
*In fact,*

$$\lim_{t\to\infty} \min_{(x,y)\in\mathcal{S}^{(m)}} \left\langle \frac{\theta_t}{\|\theta_t\|}, yx \right\rangle \Big/ \gamma_{\|\cdot\|_*^{(m)}} \to 0, \quad as\ m \to \infty. \tag{3.3}$$

Combine Theorem 3.1, Corollary 3.1 and Proposition 3.1, we conclude that the margin of the obtained solution by BPPA has non-trivial dependence on the condition number of the distance generating function $w(\cdot)$. This observation provides a strong evidence that the Bregman divergence $D_w(\cdot,\cdot)$ in BPPA is highly important to the quality of the obtained solution, and advocates a careful design of Bregman divergence when using the BPPA. Our theoretical findings also aligns the empirical evidences on the importance of divergence found in knowledge distillation and model fine-tuning (Jiang et al., 2020; Furlanello et al., 2018).

We have shown that BPPA with constant stepsize achieves a margin that is at least $\sqrt{\mu_w/L_w}$-fraction of the maximal one. Meanwhile, our complexity bound in Theorem 3.1 shows that to obtain such a margin lower bound, it might take an exponential number of iterations. We next show by employing a more aggressive stepsize scheme, we can attain the same margin lower bound in a polynomial number of iterations, while speeding up the convergence of the empirical loss $\{L_\mathcal{S}(\theta_t)\}_{t\geq 0}$ drastically.

**Theorem 3.2** (Varying Stepsize BPPA). *Given any positive sequence* $\{\alpha_t\}_{t\geq 0}$*, letting the stepsizes* $\{\eta_t\}_{t\geq 0}$ *be* $\eta_t = \frac{\alpha_t}{L_\mathcal{S}(\theta_t)}$*, then the following facts hold.*

*(1)* $\lim_{t\to\infty} L_\mathcal{S}(\theta_t) = 0$*. Specifically, for any* $t \geq 0$*, we have* $L_\mathcal{S}(\theta_{t+1}) \leq L_\mathcal{S}(\theta_t)\beta(\alpha_t)$*, where* $\beta(\alpha) = \min_{\beta\in(0,1)} \max\left\{\beta, \exp\left(-\frac{2\alpha\beta^2\gamma_{\|\cdot\|_*}^2}{L_w}\right)\right\} < 1$*.*

*(2) Letting* $\alpha_t = \frac{1}{\sqrt{t+1}}$*, we have* $\lim_{t\to\infty} \min_{i\in[n]} \left\langle \frac{\theta_t}{\|\theta_t\|}, y_i x_i \right\rangle \geq \sqrt{\frac{\mu_w}{L_w}}\gamma_{\|\cdot\|_*}$*. In particular, for any* $\epsilon \in (0, \frac{1}{2})$*, there exists a* $t_0$ *satisfying*

$$t_0 = \mathcal{O}\left(\left(\frac{L_w}{\gamma_{\|\cdot\|_*}\sqrt{\mu_w}\epsilon}\right)^8\right), \tag{3.4}$$

*such that in* $t \geq t_0$ *number of iterations, we have*

$$\left\langle \frac{\theta_t}{\|\theta_t\|}, y_i x_i \right\rangle \geq (1-\epsilon)\sqrt{\frac{L_w}{\mu_w}}\gamma_{\|\cdot\|_*}, \quad \forall i \in [n].$$

*Additionally, the convergence rate of $\{L_{\mathcal{S}}(\theta_t)\}_{t\geq 0}$ is given by $L_{\mathcal{S}}(\theta_t) = \mathcal{O}\left(\exp\left(-\frac{\gamma_{\|\cdot\|_*}^2}{L_w}\sqrt{t}\right)\right)$.*

We remark that (1) We do not optimize for the best polynomial dependence on $1/\epsilon$ in the iteration complexity (3.4), as our main goal is to show the exponential gap between the complexity presented in Theorem 3.1 and here. We refer interested readers to Appendix B, where we show that we can improve the polynomial dependence with more tailored analysis. (2) We also demonstrate that the empirical loss converges almost exponentially faster with our choice of stepsizes. (3) We reiterate that using the aggressive stepsizes does not change our established margin lower bound, and the exact convergence to the maximum margin solution demonstrated in Corollary 3.1 still holds for this scheme of stepsizes, which can be achieved with a polynomial number of iterations.

● **Inexact Implementation of BPPA.** The proximal update (2.2) requires solving a non-trivial optimization problem, and there has been fruitful results of inexact implementation of BPPA in optimization literature (Rockafellar, 1976b; Yang and Toh, 2021; Solodov and Svaiter, 2000; Monteiro and Svaiter, 2010). Here based on the *varying stepsize scheme* proposed in Theorem 3.2, we discuss the feasibility of a *gradient descent based inexact BPPA* that: (1) admits a simple implementation and achieves *polynomial complexity*, (2) retains the margin properties of exact BPPA. Specifically, at the $t$-th iteration, the gradient descent based inexact BPPA solves the proximal step

$$\widehat{\theta}_{t+1} \approx \text{argmin}_\theta\, \phi_t(\theta) := \frac{1}{n}\sum_{i=1}^n \exp\left(-\langle\theta, y_i x_i\rangle\right) + \frac{1}{2\eta_t}D_w(\theta, \widehat{\theta}_t) \tag{3.5}$$

up to a pre-specified accuracy $\delta_t$ with gradient descent. Our key observation comes from the fact that when applying gradient descent to $\phi_t(\cdot)$ with small enough stepsizes, the iterate would stay in a region that has relative smoothness $M_t$ and relative strong convexity $\mu_t$ bounded by

$$M_t \leq L_{\mathcal{S}}(\widehat{\theta}_t) + \frac{1}{\eta_t} = L_{\mathcal{S}}(\widehat{\theta}_t)\left(1 + \frac{1}{\alpha_t}\right), \quad \mu_t \geq \frac{1}{\eta_t} = \frac{L(\widehat{\theta}_t)}{\alpha_t},$$

both measured w.r.t. Bregman divergence $D_h(\cdot,\cdot)$ (Lu et al., 2018). Note that the first inequality follows by our choice of stepsize $\eta_t$ in Theorem 3.2. Thus the effective condition number $\kappa_t := M_t/\mu_t$ of $\phi_t(\cdot)$ is bounded by $\kappa_t = 1 + \alpha_t = \mathcal{O}(1)$, which implies that the $t$-th proximal step requires $\mathcal{O}\left(\kappa_t \log(\frac{1}{\delta_t})\right) = \mathcal{O}\left(\log(\frac{1}{\delta_t})\right)$ number of gradient descent steps. Summing up across $t_0$ iterations (3.4), we need up to $\mathcal{O}\left(\sum_{t=1}^{t_0}\log(\frac{1}{\delta_t})\right)$ gradient descent steps, which depends polynomially on $t_0$ even if we choose an extremely high accuracy $\delta_t = \mathcal{O}(\exp(-t))$ for each inexact proximal step (3.5).

## 4 ALGORITHMIC REGULARIZATION OF MIRROR DESCENT

Inspired by the results in the previous section, we further show that mirror descent (MD, Algorithm 2), as a generalization of gradient descent to non-euclidean geometry, possesses similar connection between the margin and Bregman divergence. We remark that our results are the first to characterize the algorithmic regularization effect of MD for classification tasks, while previous literature exclusively focus on under-determined regression problems (Gunasekar et al., 2018; Azizan and Hassibi, 2019).

---

**Algorithm 2** Mirror Descent Algorithm (MD)

---

**Input:** Distance generating function $w(\cdot)$, stepsizes $\{\eta^t\}_{t\geq 0}$, samples $\{x_i, y_i\}_{i=1}^n$.
**Initialize:** $\theta^0 \leftarrow 0$.
**for** $t = 0, \ldots$ **do**
    Compute gradient $\nabla L_{\mathcal{S}}(\theta_t) = \frac{1}{n}\sum_{i=1}^n \exp\left(-\langle\theta_t, y_i x_i\rangle\right)(-y_i x_i)$.
    Update $\theta_{t+1} = \text{argmin}_\theta\, \langle\nabla L_{\mathcal{S}}(\theta_t), \theta - \theta_t\rangle + \frac{1}{2\eta_t}D_w(\theta, \theta_t)$.
**end for**

---

**Theorem 4.1** (Constant Stepsize MD). *Let $D_{\|\cdot\|_*} = \max_{i\in[n]}\|x_i\|_*$, where $\|\cdot\|_*$ denotes the dual norm of $\|\cdot\|$, and $D_{\|\cdot\|_2} = \max_{i\in[n]}\|x_i\|_2$. Under Assumption 1, let $\mu_2$ be the strong convexity parameter of $w(\cdot)$ w.r.t. $\|\cdot\|_2$-norm. Then for any constant stepsize $\eta_t = \eta \leq \frac{\mu_2}{2D_{\|\cdot\|_2}}$, we have that*

*(1) $\lim_{t\to\infty} L_{\mathcal{S}}(\theta_t) = 0$. Specifically, we have that $L_{\mathcal{S}}(\theta_t)$ diminishes at the following rate,*

$$L_{\mathcal{S}}(\theta_t) \leq \frac{1}{\gamma_{\|\cdot\|_*}\eta t} + \frac{L_w \log^2\left(\gamma_{\|\cdot\|_*}\eta t\right)}{4\gamma_{\|\cdot\|_*}^2 \eta t} = \mathcal{O}\left(\frac{L_w \log^2\left(\gamma_{\|\cdot\|_*}\eta t\right)}{\gamma_{\|\cdot\|_*}^2 \eta t}\right).$$

*(2) We have that the margin is asymptotically lower bounded by*

$$\lim_{t\to\infty}\min_{i\in[n]}\left\langle\frac{\theta_t}{\|\theta_t\|},y_ix_i\right\rangle\geq\sqrt{\frac{\mu_w}{L_w}}\gamma_{\|\cdot\|_*}.$$

*In addition, for any $\epsilon > 0$, there exists a $t_0$ satisfying*

$$t_0=\mathcal{O}\left(\exp\left(\frac{D_{\|\cdot\|_*}^{3/2}D_{\|\cdot\|_2}L_w\eta}{\gamma_{\|\cdot\|_*}^2\mu_w^{1/2}\mu_2^{3/2}\epsilon^{3/2}}\log\left(\frac{1}{\epsilon}\right)\right)\right),\tag{4.1}$$

*such that any $t \geq t_0$, we have*

$$\left\langle\frac{\theta_t}{\|\theta_t\|},y_ix_i\right\rangle\geq(1-\epsilon)\sqrt{\frac{\mu_w}{L_w}}\gamma_{\|\cdot\|_*},$$

Theorem 4.1 shows that mirror descent attains the same $\|\cdot\|_*$-norm margin lower bound as BPPA, which is $\sqrt{\mu_w/L_w}$-fraction of the maximal margin. Note that $\mu_2 > 0$ is a direct consequence of Assumption 1 and the equivalence of norm on finite-dimensional vector space.

Similar to Corollary 3.1, let $\|\cdot\| = \|\cdot\|_A$ be the Mahalanobis distance, then MD equipped with distance generating function $w(\cdot) = \langle\cdot, A\cdot\rangle$ converges to the maximum $\|\cdot\|_*$-norm margin classifier.

**Corollary 4.1.** *Let $\|\cdot\| = \|\cdot\|_A$ for some positive definite matrix $A$. Then under the same conditions as in Theorem 4.1, the MD with distance generating function $w(\cdot) = \langle\cdot, A\cdot\rangle$ converges to the maximum $\|\cdot\|_*$-margin solution, where $\|\cdot\|_* = \|\cdot\|_{A^{-1}}$. Specifically, we have $L_{\mathcal{S}}(\theta_t)=\mathcal{O}\left(\frac{L_w\log^2(\gamma_{\|\cdot\|_*}\eta t)}{\gamma_{\|\cdot\|_*}^2\eta t}\right)$.*

*In addition, we have $\lim_{t\to\infty}\min_{i\in[n]}\left\langle\frac{\theta_t}{\|\theta_t\|},y_ix_i\right\rangle=\gamma_{\|\cdot\|_*}$. Specifically, for any given $\epsilon > 0$, there exists a $t_0$ with $t_0=\mathcal{O}\left(\exp\left(\frac{D_{\|\cdot\|_*}^{3/2}D_{\|\cdot\|_2}L_w\eta}{\gamma_{\|\cdot\|_*}^2\mu_w^{1/2}\mu_2^{3/2}\epsilon^{3/2}}\log\left(\frac{1}{\epsilon}\right)\right)\right)$, such that for $t \geq t_0$ number of iterations, we have*

$$\left\langle\frac{\theta_t}{\|\theta_t\|},y_ix_i\right\rangle\geq(1-\epsilon)\gamma_{\|\cdot\|_*},\quad\forall i\in[n].$$

*Finally, we have direction convergence that $\lim_{t\to\infty}\frac{\theta_t}{\|\theta_t\|}=u_{\|\cdot\|_*}$.*

Note that Corollary 4.1 recovers the directional convergence of steepest descent in Gunasekar et al. (2018) w.r.t $\|\cdot\|_A$, which coincides with MD with distance generating function $w(\cdot) = \langle\cdot, A\cdot\rangle$.

Theorem 4.1 guarantees the near optimal $\|\cdot\|_*$-norm margin when the distance generating function $w(\cdot)$ is well-conditioned w.r.t. $\|\cdot\|$-norm. For cases when $w(\cdot)$ is ill-conditioned, we demonstrate that there exists a class of problem for which the margin lower bound is tight.

**Proposition 4.1.** *There exists a sequence of problems $\{\mathcal{P}^{(m)}\}_{m\geq 1}$ by the same construction as in Proposition 3.1, such that the margin lower bound in Theorem 4.1 is tight up to a non-trivial factor of 2. Specifically, we have (3.2) and (3.3) also hold for MD.*

Finally, we propose a more aggressive stepsize scheme for MD that achieves the same margin lower bound. In addition, instead of requiring an exponential number of iterations (4.1) as constant stepsize MD, such a stepsize scheme only needs a polynomial number of iterations, and achieves an almost exponential speedup for the empirical loss $\{L_{\mathcal{S}}(\theta_t)\}_{t\geq 0}$.

**Theorem 4.2** (Varying Stepsize MD). *Let the stepsizes $\{\eta_t\}_{t\geq 0}$ be given by $\eta_t = \frac{\alpha_t}{L_{\mathcal{S}}(\theta_t)}$, where $\alpha_t=\min\{\frac{\mu_2}{2D_{\|\cdot\|_2}},\frac{1}{\sqrt{t+1}}\}$. Then under the same conditions as in Theorem 4.1,*

*(1) We have $\lim_{t\to\infty}\min_{i\in[n]}\left\langle\frac{\theta_t}{\|\theta_t\|},y_ix_i\right\rangle\geq\sqrt{\frac{\mu_w}{L_w}}\gamma_{\|\cdot\|_*}$. In addition, for any $\epsilon > 0$, there exists a $t_0$ satisfying $t_0=\mathcal{O}\left(\left(\frac{D_{\|\cdot\|_2}L_w}{\gamma_{\|\cdot\|_*}\mu_2\sqrt{\mu_w}\epsilon}\right)^4\right)$, such that for any $t \geq t_0$, we have*

$$\left\langle\frac{\theta_t}{\|\theta_t\|},y_ix_i\right\rangle\geq(1-\epsilon)\sqrt{\frac{\mu_w}{L_w}}\gamma_{\|\cdot\|_*},\quad\forall i\in[n].$$

*(2) We have $\lim_{t\to\infty}L_{\mathcal{S}}(\theta_t)=0$. In addition, the convergence rate is given by*

$$L_{\mathcal{S}}(\theta_t)=\mathcal{O}\left(\exp\left(-\frac{\gamma_{\|\cdot\|_*}^2}{L_w}\sqrt{t}\right)\right).$$

## 5 EXPERIMENTS

**Synthetic Data**. We take $\mathcal{S} = \{((-0.5, 1), +1), ((-0.5, -1), -1), ((-0.75, -1), -1), ((2, 1), +1)\}$. One can readily verify that the maximum $\|\cdot\|_2$-norm margin classifier is $u_{\|\cdot\|_2} = (0, 1)$. For both BPPA and MD, we take the Bregman divergence as $D_w(x, y) = \|x - y\|_2^2$, which corresponds to the vanilla proximal point algorithm and gradient descent algorithm. Note that both algorithms are guaranteed to converge in direction towards $u_{\|\cdot\|_2} = (0, 1)$, following Corollary 3.1 and 4.1.

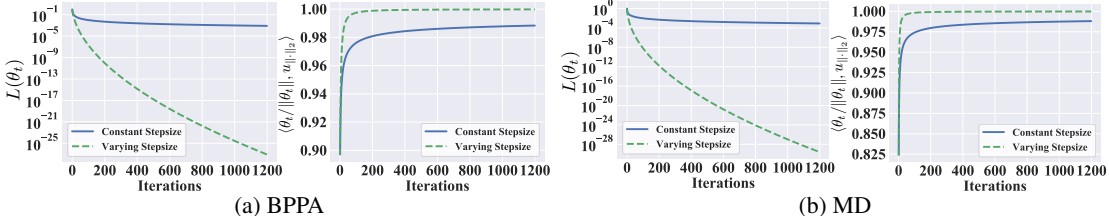

Figure 1: BPPA and MD run on the simple data set $\mathcal{S}$.

We take $\eta_t = \eta = 1$ for the constant stepsize BPPA/MD, and $\eta_t = \frac{1}{L(\theta_t)\sqrt{t+1}}$ for the varying stepsize BPPA/MD, following the stepsize choices in Theorem 3.1, 3.2, 4.1 and 4.2. To implement the proximal step in BPPA at the $t$-th iteration, we take 128 number of gradient descent steps with stepsize $0.2\eta_t$, following our discussion at the end of Section 3. We initialize all algorithms at the origin and run 1200 iterations. From Figure 1, we can clearly observe that both BPPA and MD converge in direction to the maximum $\|\cdot\|_2$-norm margin classifier $u_{\|\cdot\|_2}$, which is consistent with our theoretical findings. In addition, by adopting the varying stepsize scheme proposed in Theorem 3.2 and 4.2, both BPPA and MD converge exponentially faster than their constant stepsize counterparts.

**Data-dependent Bregman Divergence**. We illustrate through an example on how properly chosen *data-dependent divergence can lead to much improved separation* compared to data-independent divergence, even on simple linear models.

We have $n$ labeled data $\{(x_i, y_i)\}_{i=1}^m$ sampled from a mixture of sphere distribution: $y_i \sim$ Bernoulli$(1/2)$, $x_i \sim$ Unif $(\mathbb{S}_{y_i\mu}(r))$, where $\mathbb{S}_z(r)$ denotes the sphere centered at $z$ with radius $r$ in $\mathbb{R}^d$. In addition, we also have $m$ unlabeled data $\{\widetilde{x}_j\}_{j=1}^m$, following the same distribution as $\{x_i\}_{i=1}^n$, with no labels given. Clearly, the maximum $\|\cdot\|_2$-margin classifier for the mixture of sphere distribution considered here is given by the linear classifier $f^*(\cdot) = \text{sign}(\langle\cdot, \mu\rangle)$.

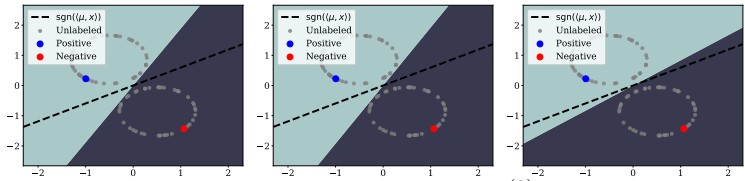

| Divergence | Alignment |
|------------|-----------|
| $D^{(1)}(\cdot, \cdot)$ | 0.8703 |
| $D^{(2)}(\cdot, \cdot)$ | 0.8175 |
| $D^{(3)}(\cdot, \cdot)$ | 0.9754 |

Figure 2: BPPA with Bregman divergence $D^{(3)}$ (right) significantly improves alignment with optimal classifier $\mu$, compared to $D^{(1)}$ (left) and $D^{(2)}$ (middle).

Table 1: $\left\langle \frac{\theta^T}{\|\theta^T\|_2}, \mu \right\rangle$ averaged over 8 runs.

We choose $n = d = 2, m = 100, r = 0.8$, and generate $\mu \sim$ Unif $(\mathbb{S}_0(1))$. We compare three types of Bregman divergence, given by $D^{(1)}(\theta, \theta') = \|\theta - \theta'\|_2^2$ (vanilla proximal point), $D^{(2)}(\theta, \theta') = (\theta - \theta')^\top \widehat{\Sigma}(\theta - \theta')$, and $D^{(3)}(\theta, \theta') = (\theta - \theta')^\top \widehat{\Sigma}^{-1}(\theta - \theta')$, where $\widehat{\Sigma} = \frac{1}{m}\sum_{j=1}^m x_j x_j^\top$ denotes the empirical covariance matrix. Note that $D^{(2)}$ and $D^{(3)}$ are data-dependent from their construction. For each divergence function, we run BPPA with 8 independent runs, the results are reported in Figure 2 and Table 1. We make two important remarks on the empirical results:

- Data-dependent divergence $D^{(3)}$ gives the best separation despite limited labeled data (in fact only 2!), much improved over data-independent squared $\ell_2$-distance $D^{(1)}$.
- Not all data-dependent divergence helps, $D^{(2)}$ shows degradation compared to $D^{(1)}$.

We further remark that by utilizing Corollary 3.1, one can completely characterize the solution obtained by BPPA for each of the divergence in closed form. Using such a characterization allows one to corroborate the empirical phenomenon with our developed theories, deferred in Appendix A.

**CIFAR-100.** We demonstrate the potential of extending our theoretical findings for linear models to practical networks, using ResNet-18 (He et al., 2016), ShuffleNetV2 (Ma et al., 2018), MobileNetV2 (Sandler et al., 2018), with CIFAR-100 dataset (Krizhevsky et al., 2009). At each iteration of BPPA, the updated model parameter $\theta_{t+1}$ is given by solving the proximal step $\theta_{t+1} = \operatorname{argmin}_\theta 1/n \sum_{i=1}^n \ell(f_\theta(x_i); y_i) + 1/(2\eta_t)D(\theta; \theta_t)$ for all $t \geq 0$, where $D$ denotes divergence function, and $\theta_0$ is obtained by standard training with SGD. We consider inexact implementation of the proximal step, discussed in (3.5). Specifically, each proximal step is solved by using SGD, with a batch size of 128, an initial learning rate of 0.1 which is subsequently divided by 5 at the 60th, 120th, and 160th epoch. We consider two divergence functions widely used in practice, defined by $D_{\mathrm{LS}}(\theta', \theta) = 1/(2n) \sum_{i=1}^n \|f_\theta(x_i) - f_{\theta'}(x_i)\|_2^2$ (Tarvainen and Valpola, 2017), and $D_{\mathrm{KL}}(\theta, \theta') = 1/(2n) \sum_{i=1}^n \mathrm{KL}\left(f_{\theta'}(x_i) \| f_\theta(x_i)\right)$ (Furlanello et al., 2018). For each of the divergence, we run BPPA with 3 proximal steps, with the proximal stepsize $\eta_t = \eta = 0.025$ for $D_{\mathrm{KL}}$, and $\eta_t = \eta = 0.2$ for $D_{\mathrm{LS}}$ ($\eta_t = 0.025$ gives significantly worse performance). For standard training with SGD, we use a batch size of 128, an initial learning rate of 0.1 further divided by 5 at the 60th, 120th, and 160th epoch. The results are reported in Figure 3.

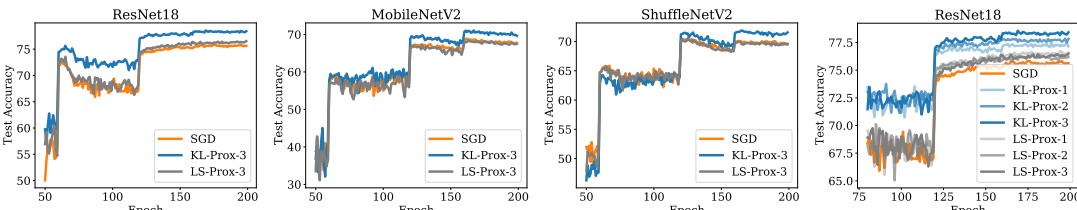

Figure 3: BPPA with divergences $D_{\mathrm{KL}}$ and $D_{\mathrm{LS}}$ on CIFAR-100 dataset. KL-Prox-$k$ denotes learning curve of the $k$-th proximal step with $D_{\mathrm{KL}}$; LS-Prox-$k$ denotes learning curve of the $k$-th proximal step with $D_{\mathrm{LS}}$.

One can clearly see from Figure 3: (1) Across different model architectures, BPPA with $D_{\mathrm{KL}}$ outperforms standard training with SGD; (2) BPPA with $D_{\mathrm{LS}}$ yields negligible differences compared to SGD. The qualitative difference of $D_{\mathrm{KL}}$ and $D_{\mathrm{LS}}$ strongly indicates that the divergence function serves an important role in affecting the model performance learned by BPPA, which we view as an important evidence showing broader applicability of our developed divergence-dependent margin theories. In addition, the learned model with $D_{\mathrm{KL}}$ improves gradually w.r.t the total number of proximal steps. For ResNet-18, the accuracy increases from 75.83% (standard training) to 78.56% after 3 proximal steps – an additional 1.4% improvement over Tf-KD$_{self}$ (see Table 2), which can be viewed as BPPA with 1 proximal step. We view such findings as the evidence suggesting the scope of algorithmic regularization associated with BPPA goes beyond simple linear models.

We make further remarks on the previously proposed method in Yuan et al. (2019), named Teacher-free Knowledge Distillation via self-training (Tf-KD$_{self}$), which is equivalent to BPPA with 1 proximal steps, using $D_{\mathrm{KL}}(\theta, \theta')$ as the divergence function. Tf-KD$_{self}$ was shown to improve over SGD for various network architectures on CIFAR-100 and Tiny-ImageNet. We include the reported results on CIFAR-100 therein in Table 2 for completeness.

| Model | SGD | Tf-KD$_{self}$ |
|---|---|---|
| MobileNetV2 | 68.38 | 70.96 **(+2.58)** |
| ShuffleNetV2 | 70.34 | 72.23 **(+1.89)** |
| ResNet18 | 75.87 | 77.10 **(+1.23)** |
| GoogLeNet | 78.72 | 80.17 **(+1.45)** |
| DenseNet121 | 79.04 | 80.26 **(+1.22)** |

Table 2: Comparison of Tf-KD$_{self}$ (2-step BPPA) and SGD on CIFAR-100.

## 6 CONCLUSION AND FUTURE DIRECTION

To conclude, we have shown that for binary classification task with linearly separable data, the Bregman proximal point algorithm and mirror descent attain a $\|\cdot\|_*$-norm margin that is closely related to the condition number of the distance generating function w.r.t. $\|\cdot\|$-norm. We list two directions worthy of future investigations. (1) Our analyses exploit the fact that the Bregman divergence is defined over the *model parameters*, while many popular data-dependent divergences are defined over the *model output* (e.g. prediction confidence). Making this non-trivial extension to data-dependent divergence can also help demystify the mechanism of the data-dependent divergence. (2) Our current analyses focus on linear models, and the extension to nonlinear neural networks requires more delicate definitions of margin and divergence. We leave this direction as our long-term investigation plan.

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
