# OpenReview forum: "Implicit Regularization of Bregman Proximal Point Algorithm and Mirror Descent on Separable Data"
_ICLR.cc/2022/Conference — ICLR 2022 Submitted_

### Official Review · Reviewer_nntq · 2021-10-30

**Correctness:** 3
**Technical Novelty And Significance:** 3
**Empirical Novelty And Significance:** 2
**Recommendation:** 6
**Confidence:** 3

**Main Review:**

## Strengths

This paper provides a framework to study theoretical guarantees of BPPA. It provides both a lower bound and an upper bound on the margin obtained by BPPA, both depending on the condition number of the distance generating function of Bregman divergence. It is a step-stone for further studies about BPPA and relevant algorithms.

## Weakness

1. The setting is restricted to binary classification on linearly separable data. I'd want to know how the results generalize to other settings, like regression and multi-classification.
2. No comparison with the convergence rate of SGD. Since SGD is the more popular alternative of BPPA, it would be helpful to state the convergence property of SGD and make a comparison with BPPA. There are some comparisons in experiments. But the difference between SGD and BPPA is not very large. It would be interesting to know which one is better in a certain situation.
3. The comparison between SGD and BPPA in the experiment with CIFAR-100 may not be a fair comparison. Tf-KD\_self is a more sophisticated method than the one studied in this paper. There are also variants of SGD, e.g. Adam, which are more commonly used and probably more efficient. So comparing vanilla SGD with Tf-KD\_self does not seem fair.


**Summary Of The Paper:**

This paper studies theoretical properties of Bregman proximal point algorithm in the linearly separable binary classification problem. The main theorem shows that the margin obtained by BPPA is lower bounded by the maximum margin multiplied by a factor, which depends on the distance generating function of the Bregman divergence. Similar results are extended to mirror descent. The theorems emphasize the importance of the choice of Bregman divergence, which is further demonstrated in several numerical experiments.

**Summary Of The Review:**

I recommend accepting the paper. Even though the setting is restricted to binary classification with linearly separable data, it adds some new understandings of the properties of BPPA and gives some suggestions on how to choose the Bregman divergence.

---

> ### Author Response · Authors · 2021-11-11
> **Response to Reviewer nntq**
>
> We deeply appreciate your positive feedback on our developed results. Here we provide our responses to your detailed remarks, and hope to address your remaining reservations.
>
> **Extensions:** We agree that extension to multiclass classification would be an interesting direction. We believe the main difficulty and the key step for multiclass classification analysis remain in establishing the connection between dual progress in BPPA/MD and the primal margin improvement, which can still be handled by our technical observations on the duality of Bregman divergence. In fact, extensions of gradient descent from binary classification to multiclass classification has been discussed in [1]. We believe combining our analysis for binary classification and the extension framework in [1], the margin lower bound we obtain in this paper can be extended to the multiclass setting. On the other hand, GD/MD for regression task has been discussed in [2]. We gently remark that analysis of over-parameterized regression problems is considered to be much easier for classification problems. By following the observation that the primal/dual iterate always falls into the data span, one can completely characterize the limit point of the GD/MD trajectory. This approach, however, is not feasible for classification tasks as there is no limit point for any sequence that minimizes the loss (iterate diverges to infinity).
>
> **Comparison to SGD:** The convergence of SGD for separable binary classification has been discussed in [3]. In particular, SGD converges to the maximal margin solution in $\ell_2$-norm. This is in contrast to MD/BPPA, where we show that the solution's margin value w.r.t. any norm $\lVert \cdot \rVert$ depends on the condition number of Bregman divergence w.r.t. $\lVert \cdot \rVert$-norm. Note that this observation serves us as one of our core messages to be delivered -- margin value has nontrivial dependence on the Bregman divergence.
>
> To achieve computational efficiency similar to SGD, one can use stochastic mirror descent, which replaces the exact gradient in Algorithm 2 with a stochastic approximation of the exact gradient. The bulk of the analysis on stochastic MD should be very similar to our current analysis, with some additional effort on controlling error propagation throughout the algorithm. Moreover, the margin properties of the stochastic MD should be the same as the current deterministic MD.
>
> Finally, we gently remark that stochastic MD strictly generalizes SGD as a broader class of optimization algorithm. The margin properties also explicitly depend on the conditionedness of Bregman divergence. BPPA can be similarly extended to stochastic BPPA, where the proximal step (2.2) is solved by a stochastic approximation algorithm, combined with our discussion on the inexact implementation of BPPA (page 6), we can also characterize the sample and iteration complexity of such stochastic version of BPPA.
>
> **Experiments on CIFAR-100:** We appreciate this suggestion on improving our empirical presentation. We gently remark that TF-KD$_{self}$ can in fact be viewed as two-step BPPA, and we will improve this empirical subsection with updated experiments with multi-step BPPA. We consider SGD mainly because several empirical and theoretical evidence shows that, despite having improved training performance over SGD, adaptive algorithms (Adagrad, Adam, etc) show worse generalization performance on various benchmark machine learning tasks [4].
>
>
>
> **References:**
>
> [1] Soudry, D., Hoffer, E., Nacson, M. S., Gunasekar, S., \& Srebro, N. (2018). The implicit bias of gradient descent on separable data. The Journal of Machine Learning Research, 19(1), 2822-2878.
>
> [2] Gunasekar, S., Lee, J., Soudry, D., \& Srebro, N. (2018, July). Characterizing implicit bias in terms of optimization geometry. In International Conference on Machine Learning (pp. 1832-1841). PMLR.
>
> [3] Nacson, M. S., Srebro, N., \& Soudry, D. (2019, April). Stochastic gradient descent on separable data: Exact convergence with a fixed learning rate. In The 22nd International Conference on Artificial Intelligence and Statistics (pp. 3051-3059). PMLR.
>
> [4] Wilson, A. C., Roelofs, R., Stern, M., Srebro, N., \& Recht, B. (2017). The marginal value of adaptive gradient methods in machine learning. arXiv preprint arXiv:1705.08292

---

### Official Review · Reviewer_Mehh · 2021-11-01

**Correctness:** 3
**Technical Novelty And Significance:** 2
**Empirical Novelty And Significance:** 1
**Recommendation:** 6
**Confidence:** 2

**Main Review:**

The typical study of implicit regularisation in the literature looks at how doing plain gradient descent on reparameterized problems lead to bias effects (such as sparsity or low rank). Moreover, these regularisation effects are because gradient dynamics along with reparameterization precisely correspond to mirror flows. I find it slightly odd that you call your setting 'implicit regularisation', because your results are dependent on a norm which is explicitly imposed by the Bregman divergence used. In this sense, I find the results obtained unsurprising. Moreover, many of the citations are related to implicit regularisation, and very few to the study of mirror descent. For example,
Orabona, Francesco, Koby Crammer, and Nicolo Cesa-Bianchi. "A generalized online mirror descent with applications to classification and regression." Machine Learning 99.3 (2015): 411-435.
Ghai, Udaya, Elad Hazan, and Yoram Singer. "Exponentiated gradient meets gradient descent." Algorithmic Learning Theory. PMLR, 2020.

I think what is missing is a review of the existing theoretical contributions on mirror descent and the authors should clarify their contributions in the context of existing works.

**Summary Of The Paper:**

This article provides an analysis of Bregman PPA / mirror descent for classification. This work is largely motivated by the recent works on implicit regularisation (of gradient descent). Their theoretical results show how the recovered margins depend on the Bregman divergence used.

**Summary Of The Review:**

In my opinion, linking this to implicit regularisation is an over-sell of the paper. The authors derive some interesting results on margin convergence, but similar results also exist in the literature, and proving convergence results in terms of the norm that you are strongly convex with respect to is not new.

---

> ### Author Response · Authors · 2021-11-11
> **Response to Reviewer Mehh, Part I**
>
> Thank you for the valuable feedback and pointers to related literature of mirror descent. Below we provide our responses to your remarks, which we hope can be helpful in addressing your current reservations.
>
> **Implicit Regularization:** We gently remark that the fundamental reason we name the studied phenomenon as implicit regularization (same as the  other works we have discussed in Page 3, the below *Observation* paragraph) is that there is absolutely no regularization imposed in the training objective. The Bregman divergence (or regularization in your context) is not added to the objective function, but instead an indispensable part of the original Bregman proximal point algorithm.
> To put it more concretely, the gradient descent update can be equivalently written as minimizing the first order approximation plus a squared distance term
> $\theta_{t+1} \leftarrow \min_{\theta}  \langle \nabla f(\theta_t), \theta \rangle + 1/(2\eta_t) \lVert \theta - \theta_t \rVert_2^2$. In this sense, gradient descent also uses a strongly convex regularization term in the update subproblem. But this is not to say that gradient descent inject a regularization term in the objective function. Such a regularization term serves merely one purpose -- to bound the update strength at each iteration. The same reasoning applies to MD (or BPPA), no regularization is added to the objective. In each update $\theta_{t+1} \leftarrow \min_{\theta}  \langle \nabla f(\theta_t), \theta \rangle + 1/(2\eta_t) B(\theta, \theta_t)$, the divergence $B(\theta, \theta_t)$ is used only to bound the strength of update from the current iterate $\theta_t$.
>
> Moreover, we understand the divergence in BPPA/MD as a part of the algorithm, and hence the results on margin lower bound (implicit regularization) is *algorithm-dependent*, instead of being divergence/regularization dependent. The gradient descent is equivalent MD with squared $\ell_2$-norm distance as the divergence, and the reason why gradient descent's implicit regularization does not have norm dependence is solely due to the fact that gradient descent only considers $\ell_2$-norm. We believe our results clearly generalize gradient descent as a special case of MD, which is a much wider class of algorithms.
>
> **Related Works:**
> * ***In the context of implicit regularization:*** Similar to our response to reviewer yiCx, our developed results and key technical observation are completely new in the literature, especially in the context of algorithmic regularization. The implicit regularization effect when using Bregman divergence for learning linear classifier has not been studied in the literature, to the best of our knowledge. The fundamental difficulty is that the update is implicitly conducted in the dual space, making it *previously unclear on how to relate the progress in the dual space to the progress of margin value defined over the primal space*. Our analysis specifically tackles this challenge by utilizing the duality of the Bregman divergence (Lemma B.3), and it is precisely this duality that produces the relationship between the margin lower bound and the conditionedness of the distance-generating function (kindly refer to our technical discussions from (B.7) through (B.10)). In fact,  to the best of our knowledge, the duality of Bregman divergence is not a property that has been extensively exploited in machine learning literature,  and *our technical observation between duality and margin progress is completely new in the literature*.
>
> * ***In the context of mirror descent***: As you have kindly mentioned, there is a fruitful line of research analyzing mirror descent for classification/regression, showing improved regret/risk convergence. We gently remark that the key conceptual difference between this line of research and the line of implicit regularization is that the latter cares more about regret/risk minimization. In fact, if one only cares about risk convergence for our task of classifying separable data, then convergence to optimal risk (zero) is much, much less interesting and challenging than the convergence of margin (kindly note that the risk convergence is provided for both BPPA and MD, constant/varying stepsize versions). This is due to the fact that there are infinitely many directions along which we can drive the risk to zero, but there also exists direction such that the margin can be arbitrarily close to zero. That is, there exists $\theta^t_{t \geq 0}$
> such that
> $
> L(\theta^t ) \to 0 ; \min_{i \in [n]} \langle y_i x_i, \theta^t/\lVert \theta^t \rVert \rangle \to 0,  t \to \infty.
> $
> Guaranteeing a  nontrivial lower bound of margin value for BPPA/MD is not something we believe can be taken for granted or expected.

---

> > ### Author Response · Authors · 2021-11-11
> > **Response to Reviewer Mehh, Part II**
> >
> >
> > **Convergence is not expected:** We gently reiterate that the objective, i.e., the training loss $L_{\mathcal{S}}(\theta)$, is not strongly convex in $\lVert \cdot \rVert$-norm. In fact, it does not even have finite solutions. The only thing that is strongly convex is the subproblem defining the update: equation (2.2) for BPPA and Algorithm 2 for MD. Note that this strong convexity says absolutely nothing about the overall convergence of BPPA and MD, they serve the sole purpose of preventing too aggressive updates.  In addition, as we have mentioned, we do not believe the convergence of margin is expected to any extent. Our *key technical observation*, on drawing the connection between *dual progress and primal margin progress via duality of Bregman divergence has never been explored in the literature*, and serves as the centerpiece of our margin analysis.

---

> > > ### Comment · Reviewer_Mehh · 2021-11-20
> > > **response to authors**
> > >
> > > Thanks for your clarifications. Implicit regularisation results in the literature are currently relating to how simple gradient descent with reparameterizations results in mirror descent and hence implicit bias towards certain solutions. I see that in your work, you directly study mirror descent, and the main contributions is that you explicitly state results in terms of classification results. I've updated my score accordingly.

---

> > > > ### Author Response · Authors · 2021-11-23
> > > > **Thank you!**
> > > >
> > > > Thanks for your timely response to our previous discussion. We really appreciate your kind acknowledgment of our previous clarifications on interpreting our results.  We gently remark that the experiment section in the updated draft has been updated, we hope that the current experimental results can deliver a clearer message that echos with our developed theories, on algorithmic regularization and the role of divergence affecting such a regularization effect.

---

### Official Review · Reviewer_yiCx · 2021-11-03

**Correctness:** 4
**Technical Novelty And Significance:** 2
**Empirical Novelty And Significance:** 2
**Recommendation:** 6
**Confidence:** 2

**Main Review:**

Strength

The paper is a very well written paper with analysis and shows the properties of the BPPA algorithm and mirror descent algorithms to learn a linear classifier using separable data. To the best of my understanding the analysis seems correct. I have gone through the proofs and they seems fine.  However, I am not sure if there is earlier work with this analysis for the particular problem they deal with.

Weakness

The main issue or concern is how relevant is the study of these algorithms to find a linear classifiers on separable data. As suggested in future work, even if they had shown empirical results on the non-linear classifiers with some heuristics would have significantly improved the contribution of the paper.

Overall my belief is that the work  has some good results but I am not sure of the relevance of these results in the present day machine learning.

**Summary Of The Paper:**

This work studies the use of Bregman Proximal Point Algorithm(BPPA) on training linear classifiers using seperable data. The paper focuses on theoretical findings of the following form

a) BPPA obtains a solution with non-trivial margin lower bound. For the mahalanobis distance, they show that the solution is a max-margin solution. They also show non-asymptotic analysis for constant step size and speed it up to exponential step size using a stepsize selection.

b) Show that the max-margin lower bound is dependent on the condition number of generating function for defining the divergence. As a result suggest that the divergence should be chosen based on the underlying space on which the data resides.

c) It shows that that above results also extend to the dual first order algorithms such as mirror descent.

**Summary Of The Review:**

The paper addresses the use of BPPA algorithm and the mirror descent algorithm to find the max-margin linear classifier in case of separable data.  My only concern is if there is some other work if this has already been addressed and relevance of the work to present day machine learning.

---

> ### Author Response · Authors · 2021-11-11
> **Response to Reviewer yiCx**
>
> We deeply appreciate your positive feedback on our developed results in this paper. Here we provide responses to your comments, which we hope can address your remaining reservations.
>
> **Related work:** The implicit regularization effect when using Bregman divergence for learning linear classifiers has not been studied in the literature. We believe the fundamental difficulty is that the update is conducted in the dual space, making it *previously unclear on how to relate the progress in the dual space to the progress of margin value defined over the primal space*. Our analysis specifically tackles this key challenge by utilizing the duality of the Bregman divergence (Lemma B.3), and it is precisely this duality that produces the relationship between the margin lower bound and the conditionedness of the distance-generating function (kindly refer to our technical discussions from (B.7) through (B.10)). In fact,  to the best of our knowledge, the duality of Bregman divergence is not a property that has been extensively exploited before in machine learning literature,  and *our technical observation between duality and margin progress is completely new in the literature*.
>
>
> **Relevance:** Our primary goal is to demonstrate the Bregman divergence has nontrivial effects on the margin value of the obtained solution.
> The developments we make in this paper characterizes explicitly how does the margin depend on the Bregman divergence through its condition number $\mu_w / L_w$.
> Studying linear models is our first step towards the general setting of practical nonlinear models, and certainly requires more technical developments beyond linear models.
> Nevertheless, we believe our results in linear models already demonstrate the close interaction between the Bregman divergence and the margin.
> We will update our experiments on the neural networks section (CIFAR-100) to demonstrate such an interaction can also lead to better performance of BPPA over SGD, for properly designed Bregman divergence.

---

> > ### Comment · Reviewer_yiCx · 2021-11-29
> > **Thank you for the clarification**
> >
> > I am now convinced with the novelty of the work as the use of dual spaces to come up with lower bounds. Previously, when I gave the score I considered this and gave the score.
> >
> > However, my concern of how relevantly this work would extend to non linear models is unknown. I ll stick to the score given for the novelty of the paper and dealing with simple setting rigorously.

---

### Official Review · Reviewer_ZTvX · 2021-11-05

**Correctness:** 3
**Technical Novelty And Significance:** 2
**Empirical Novelty And Significance:** 2
**Recommendation:** 5
**Confidence:** 4

**Main Review:**

Strength
1. Bregman proximal point is gaining interest in applications as mentioned by the authors, and understanding the property of its convergence direction (for a linearly separable data) is of significant interest.
2. This is a non-trivial extension of Gunasekar et al. (2018) showing that the GD converges in direction to the max margin solution for a linearly separable data.

Weakness
1. Applications: For a novice like me on the use of Bregman proximal point method in applications like knowledge distillation and mean-teacher learning (listed in page 1), providing such examples explicitly in the beginning with the specific choice of the Bregman distance could have been useful. I believe that this will also better motivate the presented work from the beginning, although some are mentioned later in the experiments section.
2. Non-trivial margin: It is not clear whether the convergence in direction to "non-trivial margin" solution has any important meaning. The GD converging in direction to the max margin solution somewhat suggested that the GD will generalize well, in the literatures. However, I don't see what one can claim from the non-max margin by BPPA/MD that is $\sqrt{\frac{\mu_w}{L_w}}$-times the max margin. The authors claim, after theorem 3.1, that the data-dependent divergence leads to a better separation and margin, but I don't think there is any theoretical justification for this. Let me know if I am missing something here. This also applies to the sentence "the Bregman divergence in BPPA is highly important to the quality of the obtained solution". I hope to see a more detailed explanation on the "quality" of the solution.
3. When the distance generating function is chosen to be the Mahalanobis distance, one has the max margin solution. Then, what does this make interesting over the classic GD that also find the max margin solution? The authors are aware of such fact (see the beginning of page 5), but other than this is a non-trivial derivation, I don't see any benefit of using BPPA over GD, which I expected from the abstract of this paper.
4. Conditions for the distance generating functions: To the best of my knowledge, the KL divergence in page 9 is not smooth. I suggest the authors clarify this.
5. Experiment on CIFAR-10: The authors claim that BPPA performs better than SGD in early iterations, which is not evident from the figure. In addition, the provided theory is interested in the asymptotic behavior of the methods, and asymptotically SGD and BPPA seem to behave similar in this experiment.

Comments
- page 2: I think (Bregman) proximal point algorithms are considered to be in the class of first-order algorithms.
- page 4: What is the reference point you are referring to? Could you also explain this and the corresponding non-trivial convergence results in more details?
- page 8 synthetic data: How about a constant step size that depends on $L(\theta_t)$? Is either $L(\theta_t)$ or \frac{1}{\sqrt{t+1}}$, or both making the adaptive BPPA/MD better, over the constant step BPPA/MD?
- page 8 synthetic data: How about considering the non-Euclidean distance here?
- page 8 mixture of sphere: Are the choices $D^{(2)}$ or $D^{(3)}$ standard? Any references? Are the first and second figures in figure 2 really almost the same? I personally think that the explanation of this section can be largely improved. I was not able to understand many points clearly.



**Summary Of The Paper:**

This paper extends the max margin analysis of the gradient descent (GD) method (using an exponential tail loss with linearly separable data) to its analogous analysis of the Bregman proximal point algorithm (BPPA) and mirror descent (MD). Unlike the GD converging in direction to the max margin solution, the authors show that BPPA/MD converge in direction to a solution with a margin depending on the condition number of the distance generating function.

**Summary Of The Review:**

The theory is new, but the theory does not seem to support the authors' claim that BPPA finds a better data-dependent solution. The experiment also do not seem to fully support the author's claim.

---

> ### Author Response · Authors · 2021-11-11
> **Response to Reviewer ZTvX, Part I**
>
>
> We deeply appreciate your detailed reviews and constructive feedback. Below we provide our response to your remarks and hope to address your reservations.
>
> **Applications:** Thanks for this constructive feedback. In our updated version, we will provide a concrete example on the application of Bregman proximal point method in the introduction section, to better motivate our later discussions.
>
> **Nontrivial margin:** We make three separate responses regarding this important remark.
>
> 1. As long as the distance-generating function is well conditioned, in the sense that $\mu_w/L_w = \mathcal{O}(1)$, we can obtain a margin value that is within the maximal margin value by a constant factor (e.g., 1/2). Consider the classical margin-based generalization bound,
> $R(f) \leq \hat{R}_\rho(f) +  \frac{2}{\rho} R_m(H) + \sqrt{\log (1/\delta) / m}, $
> by taking $\rho$ as the margin value, the obtained solution achieves almost the same generalization performance as the maximal margin solution, with only constant factor difference (e.g., 2). Hence the solution with a nontrivial margin lower bound yields a strong margin and *generalization guarantee that is equally well* as the maximal-margin solution when the distance-generating function is well-conditioned.
>
> 2. As an additional note, studying margin lower bound can be of independent theoretical interest.
> In fact, there has been a fruitful line of research focusing on studying the margin lower bound boosting [1,2], and a margin lower bound that is $1/2$ of the maximal margin was shown under certain conditions [1]. More importantly, we gently highlight that our analysis exploits on a key observation that has never been identified in the related literature, namely the duality of Bregman divergence. Although implicit regularization of  the mirror descent has been analyzed in the regression setting, studying on classification has not been explored to the best of our knowledge. We believe the fundamental difficulty is that the update is conducted in the dual space, making it previously unclear on *how to relate the progress in the dual space to the progress of margin value defined over the primal space*. Our analysis specifically tackles this key challenge by utilizing the duality of the Bregman divergence (Lemma B.3), and it is precisely this duality that produces the relationship between the margin lower bound and the conditionedness of the distance-generating function (kindly refer to our technical discussions from (B.7) through (B.10)). In fact,  to the best of our knowledge, the duality of Bregman divergence is not a property that has been extensively exploited before in machine learning literature,  and our technical observation between *duality and margin progress is completely new in the literature*.
>
> 3. Regarding the benefits of data-dependence Bregman divergence, we offer an alternative interpretation of Theorem 3.1. One can of course, fix a norm $\lVert \cdot \rVert$ first and then seek to find a divergence that is well conditioned w.r.t. $\lVert \cdot \rVert$. Alternatively, and  is in fact more widely used in practice, we first choose a divergence function and ask what margin properties does the solution have. Then Theorem 3.1 states that the solution has a close to optimal margin w.r.t. to a certain norm $\lVert \cdot \rVert$  as long as the distance generating function is well conditioned w.r.t $\lVert \cdot \rVert$. Note this perspective does not require fixing the norm beforehand (and in fact Bregman divergence can be well-conditioned w.r.t. multiple norms). More importantly, since the Bregman divergence function can be data-dependent (see our experiment section on the mixture of sphere distribution), the corresponding norm is also data-dependent in nature. The important implication is that such a data-dependent norm and separation can better adapt to the distribution geometry and yield better separation of the data.  We gently highlight that our experiments on the mixture of sphere distribution exactly demonstrate the importance of data-dependent divergence: the properly constructed Bregman divergence $D^{(3)}$ gives the best alignment with the optimal classifier $\mu$, leading to near-optimal separation of the two spheres. Such an improvement directly comes from the fact that $D^{(3)} \to (I_d r^2 /d + \mu \mu^\top )^{-1} $ explicitly promotes the directional component of the learned classifier $\theta^t$ along $\mathrm{span}(\mu)$. This adaptation of the learned classifier to the geometry of the mixture of sphere distributions thus gives better separation of the two spheres.
> On the other hand, another Bregman divergence $D^{(2)}$ gives even worse alignment compared to standard gradient descent (see Figure 2, Table 1). This experiment, though simple, provides clear evidence that not all data-dependent divergence can help. To achieve improved performance compared to standard training (GD/SGD),  one needs a careful design of data-dependent divergence.

---

> > ### Author Response · Authors · 2021-11-11
> > **Response to Reviewer ZTvX, Part II**
> >
> >
> > **Mahalanobis distance as Bregman divergence:**  We gently remark that we do not intend to bring forward Mahalanobis distance as an example to show the advantage of BPPA over GD (or steepest descent). The reason we introduce Mahalanobis distance is that its well-conditionedness can lead to precisely maximal margin value as a byproduct of our main Theorem 3.1. Even though steepest descent w.r.t. Mahalanobis distance also converges to maximal margin solution,   steepest descent does not show the interplay between conditionedness of Bregman divergence and the margin value. Such an interplay is precisely our core technical message, and hence the presentation of Corollary 3.1.
> >
> > **KL-divergence and NN experiments:** We agree that the settings within neural network experiments lie beyond the boundary of our established theories for linear models. Our main point of neural network experiments is to illustrate the potential of BPPA (especially with a data-dependence divergence) on more complex datasets and models. We believe the CIFAR-10 is a simple dataset, which makes the difference of BPPA and SGD not evident (as it is the case for linear models). Further experiments of multi-step BPPA on CIFAR-100 dataset will be included in our updated appendix.
> >
> > **Responses to other comments:**
> > * **First order algorithms:** We consider first-order algorithms as the class of algorithms where the iterate $\theta^t$ lies in the linear span of historical gradients $\mathrm{span}(\\{ \theta^\tau\\}_\{\tau=0\}^{t-1})$. In this sense, BPPA does not fall into this category. In fact, BPPA does not use the first-order information, instead, it uses the complete information of the objective.
> > * **Reference point**: It is quite common in the convergence of analysis that one first establishes generic convergence properties of the proposed algorithm. Such generic convergence results typically contain a reference point that has yet to be specified (e.g., see Proposition 3.1, [3]).
> > By choosing the reference point properly (typically as the optimal solution), one can specialize the generic convergence result into a concrete convergence rate.
> > The generic convergence result in our context is given in Lemma B.2, we establish the fact that
> > $
> > L(\theta_t) \leq L(\theta) + \frac{L_w}{4\eta t} \lVert \theta \rVert^2.
> > $
> > By choosing the reference point  $\theta$ properly (potentially dependent on $t$), in the sense that we can control both $L(\theta)$ and $\lVert  \theta \rVert$, then we can obtain a concrete convergence rate. Note that since infimum can only be approached asymptotically at infinity, we can not choose the ill-defined optimal solution typically done in the literature. Instead, there is a tradeoff between $L(\theta)$ and $\lVert \theta \rVert$, since we decrease $L(\theta)$ at the price of increasing $\lVert \theta \rVert$ when $\theta$ separates the data. Kindly refer to equation (B.4) on our concrete steps.
> >
> > * **Synthetic data:** (1) We believe the stepsize choice $1/[L(\theta_t) \sqrt{t+1}]$ is necessary for achieving faster convergence, while preventing aggressive updates that could lead to divergence at the same time. In particular,  using $1/\sqrt{t+1}$ leads to decreasing stepsize, which definitely leads to slower convergence than the constant stepsize version. On the other hand, using $1/L(\theta_t)$ as the step size can be too aggressive, leading to divergence when the loss is small enough.  (2) We will include a supplementary experiment of non-euclidean distance in our next version.
> >
> > * **Mixture of sphere**: Thanks for the suggestion on presentation clarity, we will improve our presentation for this section. The purpose of this subsection is to show that data-dependence Bregman divergence has nontrivial effects on the solution quality, thus echoing with our main Theorem 3.1, which shows different Bregman divergence can lead to different obtained margins. The optimal classifier for separating the mixture of sphere distribution is simply the mean $\mu$. Hence we measure the solution quality by its directional alignment with $\mu$. We compare standard gradient descent with two data dependence divergence $D^{(2)}$ and $D^{(3)}$, whose final solutions can be completely characterized by Corollary 3.1. In essence, $D^{(2)}$ simply uses the data covariance matrix to form the Mahalanobis distance, while $D^{(3)}$ uses the precision matrix. We gently emphasize that the solution obtained by using these two data-dependent regularizers is drastically different. Compared to training with standard gradient descent, training with $D^{(2)}$ has even worse directional alignment with $\mu$, while $D^{(3)}$ has much-improved alignment (see also Table 1). This observation shows that not all data-dependent divergence helps, and a careful design of divergence is necessary.

---

> > > ### Author Response · Authors · 2021-11-11
> > > **References**
> > >
> > > **References:**
> > >
> > >  [1] Rätsch, G., & Warmuth, M. K. (2002, July). Maximizing the margin with boosting. In International Conference on Computational Learning Theory (pp. 334-350). Springer, Berlin, Heidelberg.
> > >
> > >  [2] Rudin, C., Daubechies, I., Schapire, R. E., & Ron, D. (2004). The dynamics of AdaBoost: cyclic behavior and convergence of margins. Journal of Machine Learning Research, 5(10).
> > >
> > >  [3] Lan, G. (2020). First-order and Stochastic Optimization Methods for Machine Learning. Springer Nature.

---

> ### Author Response · Authors · 2021-11-23
> **Paper Revised, New experiments provided**
>
> We appreciate your constructive feedbacks provided in the reviews, and improved our paper presentation based on your detailed feedback. In addition, we have significantly strengthened our experimental part with new CIFAR-100 results. All changes were marked in blue in the updated draft. We list the key changes compared to our previous version.
>
> **Introduction:** Based on your suggestions, we have included two popular choices of divergence function used in practice to provide more context of BPPA in machine learning applications.
>
> **Experiments:**
>
> 1. **Data-dependent Bregman Divergence**: We have cleaned up the presentation for the second experiment in the experiment section. Specifically, we focus on delivering the key message that data-dependent divergence, when properly constructed, leads to much-improved data separation, which echos with our discussion right after Theorem 3.1. More technically discussions on explaining such phenomenon, by utilizing Corollary 3.1, are presented in Appendix A.
>
> 2. **CIFAR-100 Experiments**: We have conducted experiments on CIFAR-100 dataset, with ResNet-18, ShuffleNetV2, and MobileNetV2 networks. Our new empirical results strongly suggest that: (a) Construction of data-dependent divergence still matters, different divergences lead to different model quality learned by BPPA. (b) With properly constructed divergence, BPPA can improve over standard SGD training by a significant margin (an additional improvement over prior work [1]).
>
> We hope that our updated experiment section and the presentation of the paper, along with our prior responses on technical discussions (key technical innovations) can help address your remaining reservations of our paper.
>
> [1] Yuan, Li, et al. "Revisiting knowledge distillation via label smoothing regularization." Proceedings of the IEEE/CVF Conference on Computer Vision and Pattern Recognition. 2020.

---

> ### Comment · Reviewer_ZTvX · 2021-11-29
> **Thank you for the detailed response.**
>
> This is a borderline paper in my viewpoint, and it took me a while to come up with a decision. I appreciate your patience.
>
> - KL divergence without NN: KL divergence is the only non-Euclidean Bregman divergence that seems to be used practically (others were not mentioned in the paper), but this is not smooth (thus fails to satisfy assumption 1) even without neural network. (Let me know if I am wrong.) In real worlds none of the assumptions matter that much, but I think at least the assumption should satisfy for an ideal (non-NN) case. I hoped to see some discussion on the validity of assumption 1. Although being much more difficult, the analysis under a weaker condition that KL divergence satisfies could have been more interesting, at least to myself.
>
> - Second experiment with data-dependent Bregman divergence: Wouldn't the vanilla proximal point method (like GD) find the max-margin solution by the theory with the condition number 1? The authors choose how the classifier aligns with the max-margin classifier, and I am confused here, where the vanilla proximal point method fails to find the max-margin solution, unlike the Bregman proximal point methods. If we are interested in finding the max-margin classifier, why do we further need the Bregman proximal point method, over the proximal point and GD theoretically? I am not against studying the Bregman proximal point method, and I am just confused with the authors' message here. Let me know what I am misunderstanding here.
>
> - Another question on second experiment: I also asked whether the considered Bregmen divergences are standard or not, but the answer seems missing. Has it been studied by other researchers? If not, I think this section is interesting on its own, but seems a bit orthogonal to the theory given in the paper.
>
> - First-order method: This isn't that important, but for example, proximal (gradient) method, an implicit gradient method, and ADMM are considered to be first-order methods.
>
> Although the paper has some novel results, I will keep my score due to the above reasons.

---

> > ### Author Response · Authors · 2021-11-29
> > **Thank you for the response**
> >
> > Thanks for responding to our previous comments. We believe there might be some misunderstandings in our messages/results presented in the paper, we hope our response here can help us clarify these misunderstandings.
> >
> > **KL divergence without NN**: We want to gently make two remarks regarding this point:
> >
> > 1. The KL-divergence based BPPA uses KL-divergence that is defined over **prediction space**, instead of the **parameter space**.  As the input to the KL-divergence needs to be a valid probability distribution, the network’s prediction probability, instead of the network parameter, is fed into the KL-divergence. In our linear model setting, the parameter would diverge to infinity, and thus KL-divergence for the parameter is an ill-defined notion. We are explicitly tackling the case where the Bregman divergence is defined over parameters instead of prediction. As you can kindly see, in the paper conclusion section we have explicitly stated out that analyzing Bregman divergence that is defined over network’s prediction instead of on network’s parameters needs an alternative analysis, which we stated clearly as our future investigation.
> >
> > 2. We do not claim the result we obtained here is the once and for all solution on explaining the superior performance of BPPA in many applications, but we firmly believe that our analysis on the linear model itself establishes a completely new and non-trivial connection between Bregman divergence and model quality (in terms of margin). This connection we aim to emphasize also leads our response to your second concern.
> >
> > **Second experiment on data-dependent Bregman divergence**: We believe interpretations of this experiment highly correlate with interpretations of our main theory, hence we make a few gentle but conceptually important remarks regarding this point.
> >
> > 1. Vanilla proximal point and gradient descent find the maximal $\ell_2$-norm margin that fits the **training labeled data**, yet kindly note that maximal $\ell_2$-norm margin defined over the training labeled data can differ from the maximal $\ell_2$-norm margin defined over **population** (denoted by $\mu$) significantly. From Table 1, we can see that the former in fact has only 0.87 cosine similarity with the latter when labeled data is limited. Kindly also see Figure 2 (leftmost) figure for visualization, where we clearly see that maximal $\ell_2$-norm margin over the training labeled data deviates from $\mu$ significantly.
> >
> > 2. We want to highlight that BPPA with $D^\{(2)\}$ and $D^\{(3)\}$ **do not** converge to the maximal $\ell_2$-norm margin that fits the training labeled data. This is because although they have condition number as 1 (thus well-conditioned), they are well-conditioned with respect to **different norms** other than the $\ell_2$-norm. Thus, they do not converge to the maximal $\ell_2$-norm margin solution as the standard proximal point or GD. Instead, they converge to the norm for which they are well-conditioned. It is exactly this connection between Bregman divergence and norm that motivates the study of our main theorem.  If you can kindly refer to our updated Appendix A, we have provided a detailed discussion on how $D^\{(3)\}$ in particular, can boost the alignment (cosine similarity) of the learned classifier and the optimal classifier of population data ($\mu$),  by learning a maximal margin solution with respect to  a norm $\lVert \cdot \rVert$ that is data-dependent. We provide the additional $D^\{(2)\}$ divergence – which does not yield improved alignment, to further echoes with our main theorem that different Bregman divergence leads to different separation, and not all data-dependent divergence helps. In summary, our presentation of this example is to (1) show a scenario (even for simple linear models), where data-dependent divergence improves over vanilla proximal point, with detailed technical discussion (Appendix A); (2) show that not all data-dependent divergence helps, the quality of the separation highly depends on which norm does the Bregman divergence is well-conditioned with respect to.
> >
> > 3. Our main theorem explicitly stated that separation measured in a given norm depends on how well the Bregman divergence is well-conditioned with respect to its dual norm. We do not constraint ourselves to just $\ell_2$ norm as proximal point and gradient descent. As you can kindly see in Appendix A, it is this generality that allows us to show improved alignment using the properly constructed Bregman divergence $D^\{(3)\}$.

---

> > > ### Author Response · Authors · 2021-11-29
> > > **Response Cont'd**
> > >
> > > 4.  Finally, we discuss the definitions of the Bregman divergence in this example. We first remark that from definition, $D^\{(2)\}(\theta, \theta’) = \sum_\{j=1\}^m (\tilde{x}_j^\top \theta - \tilde{x}_j^\top \theta’)^2$, where $\tilde{x}_j$ is the unlabeled data. Such a divergence that regularizes the prediction difference using least-squares loss has been proposed in [1]. Definition of $D^\{(3)\}(\theta, \theta’)$ is indeed new in the literature, since the usage of precision matrix makes more sense for linear models and less so for neural networks. We additionally remark that $D^\{(3)\}(\theta, \theta’)$ also aims to regularize the prediction difference between $\theta$ and $\theta’$, the main difference from $D^\{(2)\}$ is how the prediction difference is weighted along different principal components of the unlabeled data. We are just unsure about why this experiment becomes orthogonal to our developed theory if this Bregman divergence is not popularly used in practice. We hope our previous discussions (especially point 2) can help us illustrate the main connection between this example and our main theorem.
> > >
> > >
> > > **First-order methods**:
> > > We agree that proximal gradient and ADMM are first-order methods. The first-order methods, in convention, refer to the set of algorithms that leverage first-order information (gradient/sub-gradient) at each iteration. Here BPPA clearly does not fall into the category of first-order methods, as by definition, it does not use gradient/sub-gradient information during its execution.
> > >
> > > **Reference:**
> > >
> > > [1] Tarvainen, Antti, and Harri Valpola. "Mean teachers are better role models: Weight-averaged consistency targets improve semi-supervised deep learning results." arXiv preprint arXiv:1703.01780 (2017).

---

### Author Response · Authors · 2021-11-23
**Paper revised with updated experiments**

We would like to thank all reviewers for providing constructive feedback to improve our current draft. We have made corresponding changes to the updated paper, cleaned up the presentation based on detailed comments.  We have also strengthened our experimental part with new CIFAR-100 results. All changes were marked in blue in the updated draft. We list the key changes compared to our previous version, with a focus on experiment section.


1. **Data-dependent Bregman Divergence**: We have cleaned up the presentation for the second experiment in the experiment section. Specifically, we focus on delivering the key message that data-dependent divergence, when properly constructed, leads to much-improved data separation, which echos with our discussion right after Theorem 3.1 (on data-dependent divergence leading to better separation). More technically discussions on explaining such phenomenon, by utilizing Corollary 3.1, are presented in Appendix A.

2. **CIFAR-100 Experiments**: We have conducted experiments on CIFAR-100 dataset, with ResNet-18, ShuffleNetV2, and MobileNetV2 networks. Our new empirical results strongly suggest that: (a) Construction of data-dependent divergence matters, different divergences lead to different model quality learned by BPPA. (b) With properly constructed divergence, BPPA can improve over standard SGD training by a significant margin (an additional improvement over prior work [1]). Moreover, more proximal steps can lead to further improved model quality by using BPPA.


[1] Yuan, Li, et al. "Revisiting knowledge distillation via label smoothing regularization." Proceedings of the IEEE/CVF Conference on Computer Vision and Pattern Recognition. 2020.

---

### Decision · Program_Chairs · 2022-01-20

**Decision:**

Reject

**Comment:**

The paper extends the analysis of Telgarsky (2013) and Gunasekar et al. (2018) to the Breman proximal point algorithm and to mirror descent. Upper and lower bounds show a dependency on the condition number of the distance generating function used in the Bregman divergence.

The paper received lukewarm reviews, also because the topic does not seem to be a good match for this community. In fact, none of the algorithms analyzed seem to be commonly used as optimization algorithms for deep learning, despite of the applications mentioned by the authors.

So, I didn't take into account the concerns about the relevance of the results for deep learning people, the complains about missing references from the OMD literature, and the supposed restricted setting.

However, even ignoring the above issues, the paper seems to fall squarely on the borderline. Hence, I carefully read it.

It seems to me that the analysis heavily builds on previous work, in particular the seminal paper of Telgarsky (2013) and the Fenchel-Young trick in Ji&Telgarsky (2019). The part on the Bregman divergence is novel, but technically speaking it is also straightforward for people in this sub-community. For example, Lemma B.3 is very well-known to any optimization person. Moreover, the curvature of the Bregman divergence is exactly the term one would expect to appear. So, the upper bound seems to be incremental compared to past work and it does not really add much to our understanding of this problem.
The matching lower bound is probably the only truly interesting result. However, it still does not exclude the possibility to achieve a better margin when measuring it in a different way. Indeed, measuring the margin according to the (dual) norm appearing in the strong convexity definition of Bregman divergence is not completely justified, but rather it seems a way to make the analysis work coherently.

Overall, given the overall lukewarm reviews and my evaluation of the limited novelty of the theoretical results, I recommend rejecting this paper.